https://doi.org/10.1038/s42003-021-02726-6　　OPEN
# Deep representation features from DreamDIA[XMBD] improve the analysis of data-independent acquisition proteomics

Mingxuan Gao [1,2], Wenxian Yang [3], Chenxin Li[1], Yuqing Chang[1], Yachen Liu[1,2], Qingzu He[2,4], Chuan-Qi Zhong[5], Jianwei Shuai[2,4], Rongshan Yu [1,2,3✉] & Jiahuai Han [2,5,6✉]

We developed DreamDIA[XMBD] (denoted as DreamDIA), a software suite based on a deep representation model for data-independent acquisition (DIA) data analysis. DreamDIA adopts a data-driven strategy to capture comprehensive information from elution patterns of peptides in DIA data and achieves considerable improvements on both identification and quantification performance compared with other state-of-the-art methods such as Open-SWATH, Skyline and DIA-NN. Specifically, in contrast to existing methods which use only 6 to 10 selected fragment ions from spectral libraries, DreamDIA extracts additional features from hundreds of theoretical elution profiles originated from different ions of each precursor using a deep representation network. To achieve higher coverage of target peptides without sacrificing specificity, the extracted features are further processed by nonlinear discriminative models under the framework of positive-unlabeled learning with decoy peptides as affir-mative negative controls. DreamDIA is publicly available at https://github.com/xmuyulab/DreamDIA-XMBD for high coverage and accuracy DIA data analysis.

[1] School of Informatics, Xiamen University, Xiamen, China. [2] National Institute for Data Science in Health and Medicine, Xiamen University, Xiamen, China. [3] Aginome Scientific, Xiamen, China. [4] College of Physical Science and Technology, Xiamen University, Xiamen, China. [5] State Key Laboratory of Cellular Stress Biology, School of Life Science, Xiamen University, Xiamen, China. [6] Research Unit of Cellular Stress of CAMS, School of Medicine, Xiamen University, Xiamen, China. ✉email: rsyu@xmu.edu.cn; jhan@xmu.edu.cn

Liquid chromatography coupled with tandem mass spectrometry has now become one of the most widely used approaches for high-throughput proteome data acquisition due to its capability to quantify tens of thousands of peptides per hour[1,2]. To meet the growing demand of large-scale quantitative proteome research, data-independent acquisition (DIA)[3–14] mode was established. Instead of selecting specific precursors with higher intensities for fragmentation as in data-dependent acquisition (DDA), DIA encapsulates all precursors in a pre-designed isolation range for MS2 acquisition in an unbiased way[15]. It has been proven to outperform DDA and selected reaction monitoring on various important aspects such as coverage, quantification accuracy and reproducibility[16–19].

Despite various advantages of DIA, the challenge of DIA data analysis roots in its convoluted spectra originated from signals of multiple co-fragmented precursor ions. To overcome this problem, DIA data analysis is usually performed with the peptide-centric scoring (PCS) strategy[5,20], where a spectral library that contains information of precursor ions of interest is queried against a series of raw data files to achieve higher sensitivity for large-scale complex biological samples[16,21,22]. In general, PCS software tools extract the elution profiles of the fragment ions in the library, identify and score them with a series of features and calculate the final discriminant scores for false-positive control[23–25]. Naturally, both the extracted features and the discriminative model used to generate the final scores determine, jointly, the performance of the software on protein identification and quantification. For the discriminative model, semi-supervised linear discriminant analysis[24] and semi-supervised support vector machine[26] have been used in DIA data analysis software tools. Nonlinear discriminative models such as the optional XGBoost classifier integrated in PyProphet[25] were also used. DIA-NN[27] further introduced a neural network-based model and achieved significant performance improvement. For the elution profile scoring methods, however, almost all the existing software tools use manually curated features by experts, such as Pearson correlation of the elution profiles, the shape of the elution profiles, relative intensities of the fragments, etc.[27–31], which are heuristic and may not completely cover the intrinsic characteristics of the complex elution patterns in DIA data.

Recently, deep representation learning has been extensively used for feature extraction from unstructured data[32–34] and results show that features thus obtained outperform those from conventional feature engineering methods, as deep networks are able to exploit the intrinsic joint distribution of signals in the high-dimensional feature space[35]. For DIA data analysis[36], deep learning has also been used in the prediction of fragment intensities, retention time (RT), and ion mobility[37–41], and de novo sequencing[42].

In this paper, we present DreamDIA[XMBD] (denoted as DreamDIA. XMBD refers to Xiamen Big Data which is a biomedical open software initiative in the National Institute for Data Science in Health and Medicine, Xiamen University, China), a PCS software suite for DIA data analysis using chromatogram features extracted by a deep representation network. In contrast to most existing PCS software tools[18,29,30] which only consider 6 to 10 elution profiles for peptide scoring, DreamDIA considers hundreds of additional elution profiles for each precursor including all the theoretical fragment ions, potentially unfragmented precursor ions, isotopic peaks and so forth. These elution profiles are compiled into a set of representative spectral matrices (RSMs), which are then input to a deep representation network using the Long Short-Term Memory (LSTM)[43] model to capture more informative precursor features for peptide identification.

To fully utilize the deep representation features for precursor scoring, we used a nonlinear method, the XGBoost, as the discriminative model and trained an XGBoost classifier based on all the precursors in the spectral library. Moreover, to prevent overfitting, we followed the framework of positive-unlabeled learning[44] by using the decoy precursors as affirmative negative controls in training to prevent the XGBoost classifier from picking up false targets[25,45] in the spectral library that are not detectable in a specific sample. For quantification, we calculated the weighted area under the chromatogram for each fragment ion, where the weight of each fragment is determined as the sum of its Pearson correlations with all the other fragments from the same precursor. The hypothesis behind this is that noise caused by coeluted peptides should have lower correlations with the true chromatograms.

We compared the identification and quantification performance of DreamDIA with several state-of-the-art open-source PCS tools including OpenSWATH[29], Skyline[30] and DIA-NN[27]. Our method outperformed the other tools with more target precursors identified in the two-species library test[46] and more accurate quantification in the LFQbench test[47]. Compared with the data-driven curation tool Avant-garde[31], DreamDIA presented higher quantification accuracy in the LFQbench test. Furthermore, DreamDIA could confidently identify about 1.5-fold more deamidated peptides compared with DIA-NN, and has great potential for accurate post-translational modification (PTM) profiling. DreamDIA provides a deep representation network-based feature extraction method for DIA data analysis, in combination with an interface to integrate deep learning algorithms to achieve better performance in large-scale biological and medical proteome research. The training data of the deep representation model can be easily obtained from public datasets. We also provided a trained model that can be directly applied to analyze DIA data with high coverage and accuracy, as well as an application programming interface (API) for customized model training. Users can choose our default model for data analysis from widely used data acquisition equipments conveniently without training, or use our API to train a customized model in minutes that better fits their own experiments.

## Results

**DreamDIA substantially improves the identification coverage.**
We first compared the identification performance of DreamDIA with a generic deep representation model trained on about 1 million RSMs from three public datasets with that of Open-SWATH, Skyline, and DIA-NN using the mouse cerebellum (MCB) dataset[48] (Methods). The workflow of DreamDIA and the structure of deep representation models used are shown in Fig. 1. Typically, as for routine analysis of DIA-MS data, the spectral libraries can be built from DDA master samples containing the same proteins with the corresponding DIA data, or from the DIA data directly with the aid of spectra deconvolution algorithms such as DIA-Umpire[49] and directDIA[50] in Spectronaut. In this study, we evaluated the identification performance with spectral libraries from both sources. As different software tools use different strategies for false discovery rate (FDR) estimation, it is difficult to compare the identification performance directly. Herein, we adopted the two-species spectral library method[19,27,46] for benchmarking, where the same number of Arabidopsis[51] precursors were added to the mouse sample-specific spectral libraries as false-positive controls. Proxy FDR at precursor level was calculated as the number of Arabidopsis precursors identified divided by the number of all precursors identified. Results show that DreamDIA identified more mouse precursors at different FDRs compared with other software tools on both DIA-Umpire generated library (Fig. 2a and Supplementary Data 1) and DDA master sample library

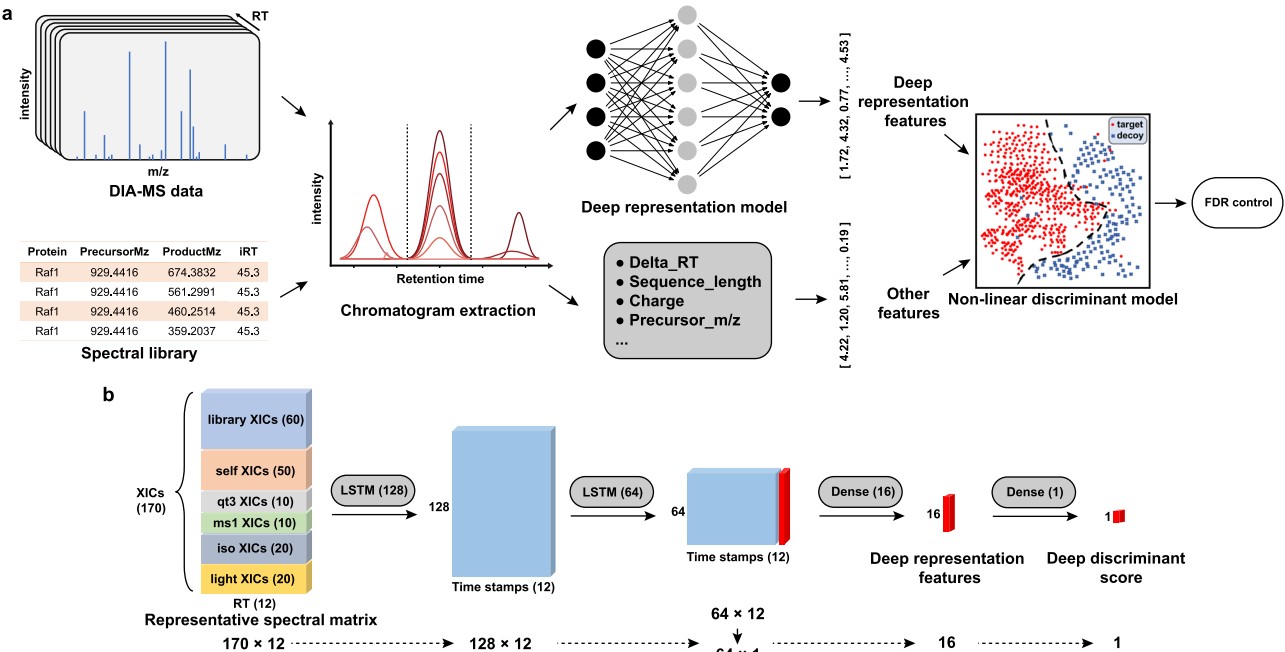

**Fig. 1 Schematic illustration of DreamDIA. a** Schematic diagram of DreamDIA. First, chromatograms of each target/decoy precursor and its corresponding fragment ions are extracted. Then the chromatograms are input to the trained deep representation model to calculate low-dimensional features. The deep representation features are further combined with other features, such as precursor m/z and charge, to calculate the final discriminant score. **b** The deep representation model in DreamDIA. The input RSM consists of six types of XICs, namely, *library*, *self*, *qt3*, *ms1*, *iso*, and *light* XICs. The model contains two LSTM layers and two full-connected (FC) layers, through which the input RSM was transformed to the 16-dimensional deep representation features by the first FC layer and to the deep discriminant score (*dds*) by the second FC layer.

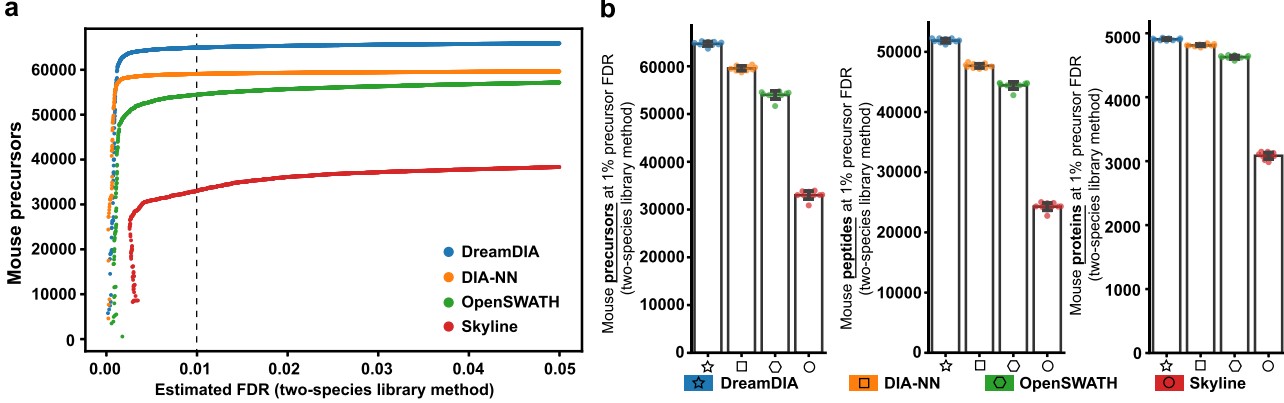

**Fig. 2 Identification performance evaluation of DreamDIA on the mouse cerebellum dataset with two-species library method. a** Identification performance on the S1-1 run of the MCB dataset. The numbers of mouse precursors identified at different FDRs were plotted. Each point stands for an *Arabidopsis* (false-positive) precursor and its discriminant score as a cut-off value. The x-axis value stands for the estimated FDR, calculated as the number of *Arabidopsis* precursor with higher discriminant score than this cut-off value divided by the number of all the precursors with higher discriminant score than this cut-off value. The y-axis value stands for the number of mouse precursors with higher discriminant score than this cut-off value. **b** Identification performance on all 10 samples of the MCB dataset. The numbers of mouse precursors, peptides, and proteins at 1% precursor FDR (the respective numbers indicated by the dashed line in (**a**)) were plotted. Each error bar stands for the mean and standard deviation of the results of $n = 10$ biologically independent runs.

(Supplementary Fig. 1a). Further evaluation of all 10 runs on the MCB dataset shows that DreamDIA identified more mouse precursors, peptides, and proteins at 1% precursor FDR using both libraries (Fig. 2b, Supplementary Data 2 and Supplementary Fig. 1b).

To fully understand the mechanism behind this performance improvement, we extracted the RSMs of the mouse precursors reported by DreamDIA that were missed by other tools. Manual inspection on these elution profiles (example shown in Supplementary Fig. 2) shows that in most of these cases, the elution profiles of the ions listed in the spectral libraries did not contain sufficient information for peptide identification. Hence, traditional approaches that only examined these elution profiles could not achieve satisfactory results. On the other hand, as DreamDIA examined significantly more theoretical ions with its deep representation network, these target peptides can be successfully recovered based on the confluent elution profiles from the additionally examined ions.

Interestingly, in accordance with previous work from Parker et al.[52], which found that libraries from DIA-Umpire have better data completeness compared with DDA master sample libraries, DreamDIA identified significantly more precursors when it was used together with DIA-Umpire library than with DDA master sample library (Fig. 2 and Supplementary Fig. 1). Of note, although using the library constructed with DDA run on master samples is still a preferred approach for DIA data analysis, the identification performance of such approach may be subject to the discrepancies in signal characters and RT in-between DDA and DIA runs[31]. Thus, this result reveals the potential of using libraries produced directly from deconvolution of DIA data to not only remove the dependency on sample-specific complementary DDA runs, but also improve the identification coverage by using it together with improved peptide scoring strategies.

To test DreamDIA for analyzing data acquired at different gradient lengths, we further used the HeLa dataset[27,46] for comparison. As shown in Supplementary Fig. 3, DreamDIA still achieved the best identification performance compared with the other software tools for data acquired at all four different gradient lengths ranging from 0.5 to 4 h.

In addition, we also evaluated whether DreamDIA could correctly identify the deamidated peptides with extremely small mass shift of 0.9840 Da on the MCB dataset (see Supplementary Note 1 and Methods). With FDR calculated by the two-species method, DreamDIA confidently identified about 1.5-fold more deamidated peptides compared with DIA-NN (Supplementary Fig. 4), indicating the improved capability of DreamDIA to identify PTM peptides with small mass shifts.

**DreamDIA produces reliable identification improvements**. To test the reliability of the identification improvement brought by DreamDIA, we compared the identification consistency between DreamDIA and the other software tools. Precursors, peptides, and proteins identified at least once by each software tool at 1% precursor FDR in all 10 parallel runs on the MCB dataset were considered. Results show that DreamDIA has highly consistent precursor, peptide, and protein identifications as well as steadily more unique identifications compared with OpenSWATH, Skyline, and DIA-NN (Fig. 3a–c and Supplementary Data 3). The results from decoy-based default FDR method (Fig. 3d–f and Supplementary Data 3) also show comparable consistency with the ones from two-species FDR method (Fig. 3a–c), which indicates that DreamDIA can produce highly consistent identification results when decoy-based FDR estimation is used. Furthermore, the Gaussian-like logarithmic intensity distribution of the identified precursors (Fig. 3g, h and Supplementary Data 3) indicates that DreamDIA has no abundance bias for peptide identification. Finally, the MS1-related subscores calculated by OpenSWATH further confirmed that the majority of those additionally identified precursors by DreamDIA indeed had MS1 signals, and the distributions of these subscores were very similar to those of consensus precursors identified by all the software tools (Supplementary Fig. 5).

**Impact of representative spectral matrix design to identification performance**. To improve the identification performance, we expected to involve as many signals a peptide can induce during mass spectrometry characterization as possible so that the deep representation model has sufficient information to discriminate between the elution patterns of real peptides and those of decoys. To this end, in DreamDIA, six different types of fragment ion XICs were collected and stored in the RSM, which is then input to the deep representation models for peptide identification (Fig. 1b, Methods). The *library* part contains XICs of the

fragment ions in the spectral library at three different resolutions, more specifically, $r$, $0.2 \cdot r$, and $0.45 \cdot r$, where $r$ denotes the basic resolution that can be specified by users according to the acquisition resolution. These XICs reflect elution information of the most significant fragments at the library building stage. On top of the *library* part, we also considered XICs of the $(M + 1)/q$ isotopic peaks of the corresponding *library* fragments as the *iso* part, where $M$ and $q$ stand for the mass and charge of the fragment ion respectively. In addition, XIC at $(M - 1)/q$ of each *library* fragment is also included as the *light* part indicating the possibility of this fragment being actually a heavy isotopologue of a light fragment. The *self* part contains XICs of all the theoretical fragment ions with 1 and 2 charge(s) for one precursor. Besides fragment ions, we put XICs of precursor ion with the same three resolutions and its $(M + 1)/q$ to $(M + 4)/q$ isotopic peaks as well as the XIC at $(M - 1)/q$ at the basic resolution into the *ms1* part. XICs of unfragmented precursor ion and its $(M + 1)/q$ to $(M + 4)/q$ isotopic peaks were also considered as the *qt3* part. These two parts contain elution information of the precursor ion in both MS1 and MS2 spectra. Note that except for the XICs of the *library* part, most elution profiles above were not considered in conventional algorithms for peptide identification and quantification.

To test whether these additional elution profiles contribute to peptide identification, we trained eight deep representation models with ablation of different XIC part, and compared their performance on the MCB dataset with both DIA-Umpire library and DDA master sample library. All the models had the same architecture and hyperparameters except for the input dimension, and were trained on the same training set. As expected, removing any XIC part from RSM led to a performance drop in mouse precursors identification results (Fig. 4a, b and Supplementary Data 4, 5), confirming the contribution of all the XIC parts included in our design. Top performance drops were observed when the *library* XICs with $0.2 \cdot r$ and the *ms1* XICs were excluded, indicating the importance of the information embedded in these XIC parts to the identification tasks. We also evaluated deep representation models with different hyperparameters, and small variations were observed in a relatively wide parameter space (Supplementary Table 1).

To further illustrate the contributions of each XIC part to the deep representation model, we used the SHAP[53] deep explainer to visualize the feature importance distribution of the RSM. SHAP is a unified framework based on additive feature attribution methods for interpreting predictions of complex models by assigning each feature an importance value for a particular result. Herein, we randomly picked 10,000 RSMs from the training sets, with 5402 RSMs from real peptide precursors and the rest from decoys. The heatmaps of the averaged SHAP values of these RSMs confirm the contribution of each XIC part to the deep representation model (Supplementary Fig. 6), with the most significant contributions from *library*, *self* and *ms1* parts. As expected, the SHAP values of decoys had an opposite sign to those from real peptide precursors, and so did the SHAP values of the *light* part to those of other parts, indicating their opposite effects on the peptide identification results. Moreover, the signals in the middle of the acquisition cycles, where they peak, show higher feature importance than those at both sides of the scanning window.

**Nonlinear discriminative model improves the classification performance**. The final discriminant score of each precursor in DreamDIA was calculated by a binary classifier to incorporate additional features not captured by the deep representation model (Supplementary Note 2) for better identification results (Fig. 1a).

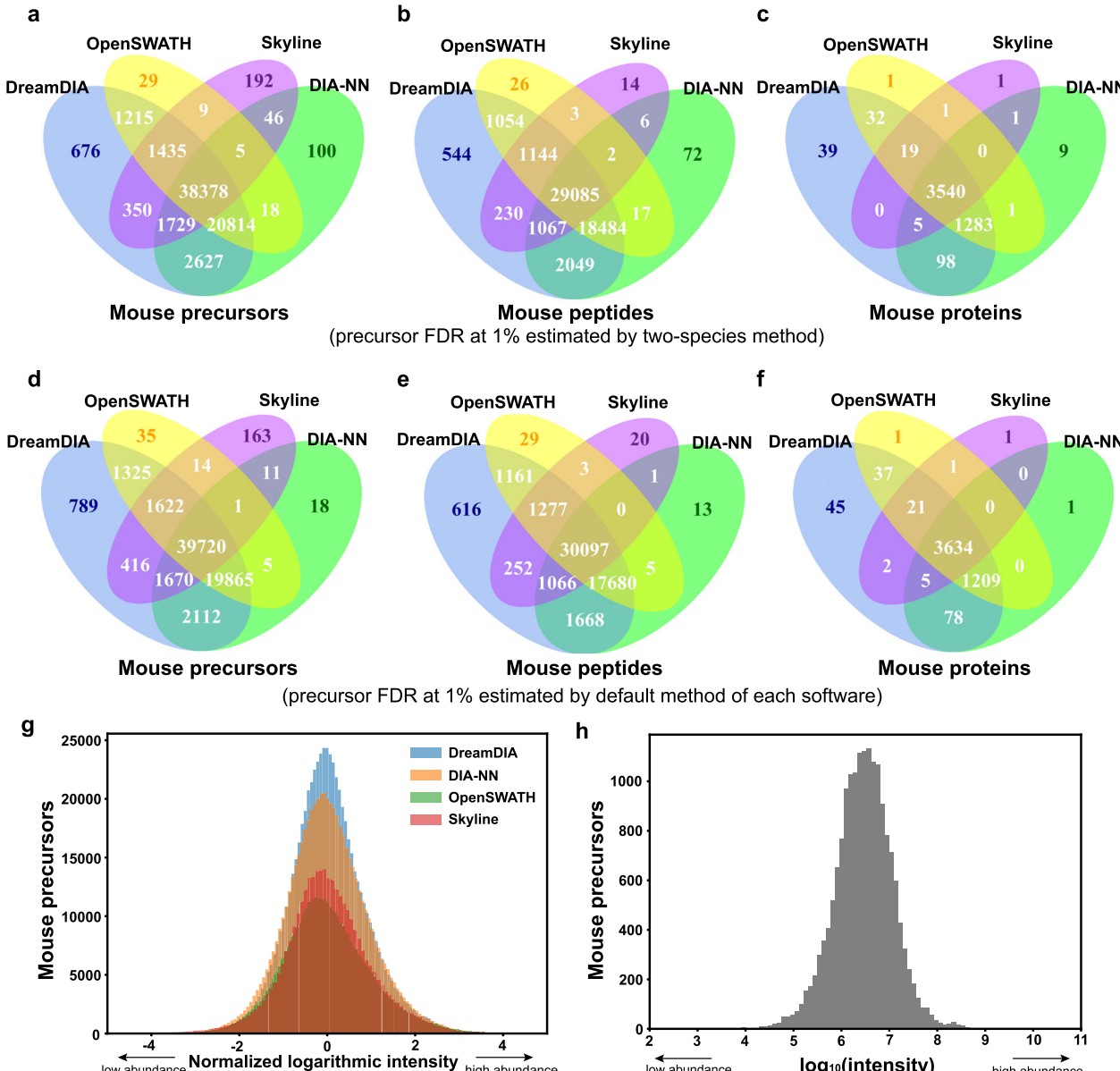

**Fig. 3 Identification results evaluation of DreamDIA on the mouse cerebellum dataset. a–f** Identification consistency of mouse precursors, peptides, and proteins at 1% precursor FDR using two-species library method and decoy-based default FDR method. Precursors identified at least once for each software tool in all 10 runs on the MCB dataset were considered. **g** Logarithmic intensity distributions of the mouse precursors identified at 1% precursor FDR using two-species library method. All the precursor identification records in all 10 runs for each software tool were considered. To make the distributions comparable, the logarithmic intensities were z-normalized by subtracting the mean intensity and then divided by the standard deviation. **h** Log$_{10}$-intensities of the precursors uniquely identified by DreamDIA in all the biologically independent 10 runs on the MCB datasets.

Similar to other DIA data analysis tools, the training of the final discriminative model was performed using a positive-unlabeled learning framework based on labels provided by the spectral library, where the decoy precursors produced by in silico decoy generation methods were taken as confirmed negative control, and the target precursors were treated as unlabeled class as they may also include negative precursors. DreamDIA integrated five widely used decoy generation methods (Supplementary Fig. 7a) and took the shuffle algorithm as the default option as suggested by Röst et al.[29]. We tested seven commonly used classifiers including LDA, logistic regression, CART decision tree, Adaboost, gradient boosting decision tree (GBDT), random forest and XGBoost on the MCB dataset. The deep discriminant score (*dds*), which is calculated by the deep representation model to indicate the probability that each precursor in the spectral library belongs to a

real peptide (Methods), was also included in our comparison. With the features extracted by the deep representation network, all the classifiers showed relatively better identification performance compared with other PCS software tools (Supplementary Fig. 8). Interestingly, although the identification performance of *dds* was inferior to those of other classifiers as it only considers the features from the elution profile, it still delivered significantly better performance compared with other traditional methods where more information such as the length and charge of peptides were considered, signifying the importance of fully utilizing the information embedded in elution profiles for peptide identification in DIA. Among all the tested classifiers, tree-based ensemble models including GBDT, XGBoost, and random forest obtained better performance, while linear models such as logistic regression and LDA performed slightly worse than nonlinear models.

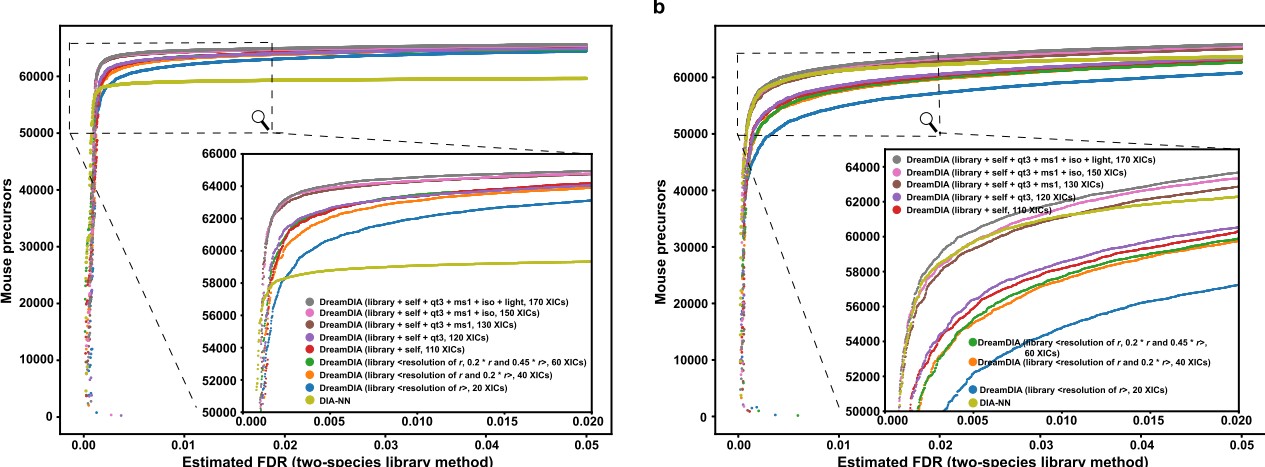

**Fig. 4 Optimization of the representative spectral matrix compositions in DreamDIA.** The deep representation models trained on the RSMs with eight different compositions as well as DIA-NN were tested on the S1-1 run of mouse cerebellum dataset using **a** DIA-Umpire library and **b** DDA master sample library. Two-species FDR estimation method was used as in Fig. 2. The numbers of mouse precursors identified at different FDRs were plotted. *r*, the basic resolution that can be specified by users according to the acquisition resolution.

t-SNE dimension reduction embedding of the 16-dimensional deep representation features (Supplementary Fig. 9) shows that although the positive and negative precursors were highly distinguishable in the feature space, the separation boundaries between them can be highly nonlinear. Hence, it could be difficult for linear classifiers to fully discriminate between real peptide signals and noise.

Finally, we performed a comprehensive validation of all the compatible decoy generation methods of DreamDIA, DIA-NN, OpenSWATH, and Skyline on the MCB dataset (Supplementary Fig. 7b). All the methods presented highly stable performance for peptide identification except the random mass shift algorithm in Skyline that induced a substantial drop.

**DreamDIA shows better quantification performance.** The *library* XICs in RSM reflect the elution profiles of the most significant fragment ions during library building, and show higher feature importance in the experiments above. Thus, we used the areas of these XICs weighted by the sums of their Pearson correlations with all the other XICs for precursor quantification (Methods). We compared the quantification performance of DreamDIA, OpenSWATH, and DIA-NN using the LFQbench software suite[47], and evaluated the accuracy of quantification algorithms by measuring their abilities to recover the groundtruth abundance ratios on its internal dataset containing proteins from different species (human, yeast and *E. coli*) at known ratios. We tested with the HYE110 and the HYE124 samples with 64-variational-window setup acquired from TripleTOF 6600 systems. Each software tool outputs results at 1% precursor FDR based on its own standard. Overall, DreamDIA achieved more accurate quantification performance at both peptide and protein levels as reflected by smaller deviations from its recovered ratios to the groundtruth compared with the other two tools on both HYE110 (Supplementary Fig. 10) and HYE124 (Supplementary Fig. 11) datasets. The global accuracy metric calculated by the LFQbench package, which reflects the median deviation of the calculated log-ratios of the quantification results to the expected values, further confirmed the superior quantification performance of DreamDIA at both peptide and protein levels compared with the other two tools on both datasets (Supplementary Tables 2, 3).

We further tested whether the inclusion of more fragments would affect the quantification accuracy. We modified the original quantification calculation to include top 3 to 15 *self*

fragments sorted by their Pearson correlations with the first *library* fragment, and their Pearson correlation weighted areas were added to the final results. Inclusion of these fragments has relatively small impact on the quantification results, with slight improvement on yeast peptide and protein quantification but slight degradation on *E. coli* peptide and protein quantification in terms of deviation of the measured ratios to the groundtruth (Supplementary Figs. 12, 13 and Supplementary Table 4). It is noteworthy that although the deep representation model can extract useful information from the RSMs for precursor identification, many theoretical XICs may contain complex interference in their elution profile waveforms. Therefore, inclusion of XICs with poor signal-to-noise ratio without performing sophisticated denoising may not improve quantification results.

**Benchmarking of DreamDIA with Avant-garde.** Avant-garde[31] is a recently published data curation tool integrated in Skyline for DIA and parallel reaction monitoring (PRM) analysis. After routine analysis performed by Skyline, it can select the best XICs using genetic algorithms, adjust peak boundaries and provide intuitive subscores. With the refinement of Avant-garde, fragments with interference are discarded and a high quantification precision can be achieved. In this work, we compared DreamDIA with Avant-garde using the LFQbench HYE110 dataset, which was also used in the original paper of Avant-garde[31]. Skyline was included for comparison since it could be regarded as an unoptimized version of Avant-garde. As expected, DreamDIA could identify more peptides than both Skyline and Avant-garde (Supplementary Fig. 14a). For quantification, DreamDIA also achieved higher accuracy as indicated by its closer medians of the calculated log-ratios to the expected values compared with Skyline and Avant-garde, while Avant-garde showed lower quantification variation than DreamDIA due to its relatively aggressive fragment filtering operation (Supplementary Fig. 14b–d).

**Computational performance.** The running time of DreamDIA is proportional to the number of acquisition cycles specified for each precursor to be analyzed by the deep representation model. Overall, DreamDIA (v2.0.2) is about two-fold slower than OpenSWATH when 50 acquisition cycles are specified, and about seven-fold slower than OpenSWATH for 500 acquisition cycles (Supplementary Fig. 15). DreamDIA was implemented in Python

which is more suitable for feasibility validation. It is envisioned that the computational complexity of DreamDIA could be further reduced by reimplementing with more efficient programming language such as C++ that was used in DIA-NN and Open-SWATH, or incorporating algorithmic improvement such as introducing pre-filtering algorithms to reduce the numbers of RSMs for analysis in future developments.

## Discussion

Feature extraction of elution profiles is crucial for both accurate identification and quantification of peptides and proteins in DIA data analysis. During the past decades, researchers have developed dozens of feature extraction rules to depict the elution patterns of peptides in DIA data and suggested on selecting the scoring methods for different projects. However, as the distributions of the signals and noise from different samples and equipments can be highly variable, it is difficult for manually curated scores to capture all such variance. Moreover, the common approach adopted by most current DIA data analysis softwares, where only fragment ions from the spectral library are examined, imposes another important limitation to the identification performance as shown in our results.

With the fast accumulation of large amounts of DIA data being generated by consortia around the world in recent years, data-driven approaches are preferable to produce more accurate and robust results without interference from subjective opinions. We developed DreamDIA, a deep neural network-based DIA data analysis software. By adopting a data-driven approach, the deep representation network used in DreamDIA learns the distribution of much broader elution profiles generated by DIA, from which a more comprehensive representation of elution patterns can be derived. As a result, DreamDIA prominently improves the identification and quantification performance at both peptide and protein levels compared with traditional approaches where only a small number of ions selected by manually curated rules are considered. Of note, one important characteristic of data-driven approaches is that they can solve complex machine learning problems by simple statistical models supplied with all relevant data available rather than depending on customized features derived from domain knowledge[54]. However, it can be anticipated that the performance of such a model will be highly associated with the training data size. Indeed, in many data-driven machine learning tasks such as object recognition and natural language processing, there is a clear association of increased benchmark performance with an increasing amount of training data[55]. In this perspective, it is anticipated that the performance of DreamDIA could further improve with the availability of more high-quality training data in near future.

To make DreamDIA widely available to users of all backgrounds to process their DIA data, we implemented a complete DIA data analysis pipeline from raw data to quantification results as a standalone software package. We anticipate that with the significantly improved identification coverage, fidelity, and producibility, DreamDIA will facilitate the integration of the large volumes of DIA data currently being generated around the world.

## Methods

**Deep representation model in DreamDIA**. The key step of DreamDIA is to extract relevant features of the chromatograms with a deep representation model. The input to the deep representation model is the RSM (Fig. 1b), a matrix consisting of 170 XICs across six types of elution profiles, namely, *library, self, qt3, ms1, iso,* and *light*. The *library* part contains 60 XICs in total, which include fragment ions in the spectral library at three different resolutions, $r$, $0.2 \cdot r$ and $0.45 \cdot r$, where $r$ denotes the basic resolution in ppm or Da. For each resolution, 20 XICs with highest intensities are kept. Zero-filling is used if <20 fragments are available in the library. After extraction, the XICs at the basic resolution are sorted by the sum of their Pearson correlations with all the other *library* XICs. Once the order of XICs at

the basic resolution is fixed, XICs at the other two resolutions are sorted according to their corresponding relationships to the XICs at the basic resolution. The *iso* part contains 20 XICs at $(M+1)/q$ of each *library* fragment, which corresponds to potential isotopic peaks of these fragments. In addition, XIC at $(M-1)/q$ of each *library* fragment, which indicates the possibility of this fragment being actually a heavy isotopologue of a light fragment, is also included as the *light* part of RSM. The *self* part contains XICs of all the theoretical fragment ions from one precursor. Specifically, for precursors with two charges, fragment ions with one charge are considered. For precursors with charges greater than two, fragment ions with one and two charge(s) are considered. Intensity-based filtering or zero filling is used when more or less than 50 fragments are available. After extraction, the *self* XICs are sorted by their Pearson correlations with the first *library* XIC. Besides fragment ions, we also include XICs of precursor ion at the three aforementioned resolutions, its $(M+1)/q$ to $(M+4)/q$ isotopic peaks and the XIC at $(M-1)/q$, both at the basic resolution into the *ms1* part. XICs of unfragmented precursor ion and its $(M+1)/q$ to $(M+4)/q$ isotopic peaks are also considered as the *qt3* part. The XICs in these two parts are in a fixed order for all RSMs. The RT width of the RSM is set to 12 acquisition cycles (around 3 s per cycle for most equipments) by default, which is long enough for most elution signals as we manually verified on several DIA datasets.

Our deep representation network consists of two LSTM layers and two full-connected layers. The first LSTM layer has 128 neurons with an input dropout rate of 0.4 and a recurrent dropout rate of 0.3. The second LSTM layer with 64 neurons and the same dropout settings was then stacked on the first layer. Two full-connected layers with 16 and 1 neuron(s) respectively were added on the top of the model. The rectified linear unit (ReLU) activation function was used for hidden layers, while the sigmoid function was used for the final layer to obtain an output ranging from 0 to 1. The model was trained on the RSMs in the training datasets to differentiate real precursors from decoys with a cross-entropy loss function. The output of the final layer of the trained deep representation model, the *dds*, represents the likelihood that a certain RSM is from a target peptide present in the sample and is used for RT normalization and peak picking. In addition, the 16-dimensional output of the second to last layer of the model is used as the deep representation features, which is further input to the discriminative model to generate the final discriminant score for each precursor from the spectral library (Fig. 1b).

In our experiments, the training data of the default deep representation model for DreamDIA (v2.0.2) includes three public datasets, the HEK293 dataset[56], the L929 mouse dataset[57], and the BioIDS-OT dataset[58] (Supplementary Table 5). For the HEK293 dataset and the L929 mouse dataset, DIA-Umpire libraries were used, while for the BioIDS-OT dataset, DDA master sample library was used. Decoys were generated using the DreamDIA decoy generation module. Subsequently, DIA-NN, OpenSWATH, and Pyprophet[25] were used to find RT of each target or decoy precursor in the spectral libraries across all runs. Results with FDR <1% were retained. All the necessary XICs were extracted and saved as RSMs. In total, we obtained around 1 million RSMs from the three datasets, which were split by 7:3 as training and validation data for the deep representation model. The validation loss stopped decreasing after 11 epochs. We then retrain the model with all the 1 million RSMs for 11 epochs to obtain the final model. Model training was performed on a GeForce GTX 1080 GPU.

**Building sample-specific spectral libraries**. We tested DreamDIA with two types of sample-specific spectral libraries in our experiments. One is the DDA master sample library as provided by the original datasets and the other is the library generated by the spectra deconvolution algorithm DIA-Umpire[49]. The following workflow was used to create the DIA-Umpire libraries. First, we transformed the raw data files to centroided mzXML files by ProteoWizard[59] (v3.0.19317) with all the other arguments unchecked. The resulting mzXML files were then processed by DIA-Umpire (v2.0) to generate pseudo MS/MS spectra. X!Tandem[60] and Comet[61] were used to search these pseudo spectra, and the searching results were further filtered by PeptideProphet[62] and ProteinProphet[63] in TPP[64,65] (v5.2.0). Finally, the sample-specific spectral libraries were generated by SpectraST[66].

**DreamDIA workflow**. DreamDIA supports centroided mzML or mzXML MS data files as input. It has also integrated the cross-platform MS file conversion tool ThermoRawFileParser[67] to read raw files directly from Thermo Fisher equipments. After reading a spectral library, the decoy generation module of DreamDIA generates a decoy for each precursor ion. Five commonly used decoy generation methods including *shuffle, reverse, pseudo-reverse, mutate* and *shift* are integrated in DreamDIA as options (Supplementary Fig. 7a). Decoys generated from other software tools such as OpenSWATH can also be used directly. After acquiring the inputs file, a set of endogenous precursors in the library was randomly sampled for RT normalization. The best RSMs were identified based on the *dds* output from the deep representation model for each precursor in the sampled set across the whole RT gradient. Then a regression model was fit for the time points of the best RSMs against their normalized RT, by which the RTs of the other precursors in the spectral library can be predicted. Herein, users can choose either a linear or a nonlinear model in DreamDIA according to the RT distributions of the specific datasets. For higher robustness, the Random Sample Consensus (RANSAC) algorithm[68] and the locally weighted scatterplot smoothing (LOWESS) algorithm

are used in the linear model and the nonlinear model respectively to detect outliers. To distinguish between target and decoy precursors, a nonlinear discriminative model was used in DreamDIA based on the deep representation features in combination with other features such as peptide length and charges (Supplementary Note 2). Due to the existence of false targets in a spectral library, this model is trained based on the principle of positive-unlabeled learning[44] with decoy peptides as affirmative negative controls while treating targets as unlabeled. We adopted the naive approach that training a binary classifier directly between confirmed decoys and undetermined target precursors to estimate the probability for an RSM of belonging to a real peptide. XGBoost was chosen as the default discriminative model for its superior performance (Results) and high efficiency. The software also provides an option for users to choose other types of classifiers such as random forest as the discriminative model. To further reduce the bias of RT dislocation between spectral libraries and DIA runs, we adopt the test-time augmentation strategy where all the RSMs within a certain range centered at the predicted RT with *dds* higher than a given threshold were included as potential targets for a precursor in the positive-unlabeled learning process. All the features of candidate RSMs are used to train a binary classifier with depths of 6 and 12 for XGBoost and random forest models, respectively, to assign a discriminant score for each RSM. The number of estimators in the random forest model is set to 200 by default.

Finally, the discriminant score for each RSM was calculated with the trained model, and the RSM with the highest discriminant score for each precursor is kept for FDR control. The FDR is estimated by dividing the number of target precursors by the number of all precursors with discriminant scores exceeding a cut-off score. For a specific FDR level, DreamDIA searches for a proper cut-off score for valid identifications. Protein-level FDR is calculated similarly by dividing the number of target precursors by the number of all precursors exceeding a cut-off score for each protein, respectively.

**Peptide and protein quantification**. For peptide quantification, a weighted area method was used to mitigate the noise brought by other co-eluting ions.

$$Q(Precursor_k) = \sum_{i=1}^{n} \sum_{j=1}^{n} Corr(C_{k,i}(t), C_{k,j}(t)) \cdot \int_{t_0}^{t_E} C_{k,i}(t)\mathrm{d}t \quad (1)$$

Here $C(t)$ refers to the elution chromatogram of an ion. *Corr()* denotes the Pearson correlation of two fragment ions, and the integration calculates the area under the ion's chromatogram. This method quantifies a precursor by the areas of the chromatograms of its fragment ions, and the weight of each fragment is set to the sum of its Pearson correlations with all the other fragments from the same precursor. In general, the top six *library* fragments associated with each precursor sorted by the sum of their Pearson correlations with all the other *library* XICs were used for quantification. For protein-level quantification, the sum of the intensities of the top three abundant precursors for each protein was calculated.

**Benchmarking of precursor identification**. We compared DreamDIA (v2.0.2), OpenSWATH[29] (v2.6.0), Skyline-daily[30] (v21.0.9.118) and DIA-NN[27] (v1.7.11) in terms of precursor identification. For DreamDIA, default settings were used in all experiments except that "--n_cycles 50" was specified for the MCB dataset with DDA master sample library. For DIA-NN, the Linux command-line tool with default settings was used. OpenSWATH was run with options "-readOptions cacheWorkingInMemory -batchSize 0 -rt_extraction_window 1200 -threads 20". Then the output was processed by PyProphet-cli (v0.0.19)[25] with "--lambda=0.4 --statistics-mode=local" options. Suboptimal peak groups were subsequently discarded. For Skyline, the step-by-step settings are described in Supplementary Note 3. We did not perform extensive parameter optimization for Skyline and OpenSWATH, as it was shown that DIA-NN outperforms both of them[27], which is also validated by our results.

To build the two-species spectral library for benchmarking, the proteins from other different species were added to the sample-specific spectral libraries as target precursors, and were used to evaluate the false-positive identifications. The MCB dataset[48] and the HeLa dataset[46] were first processed by DIA-Umpire to produce the sample-specific libraries. The DDA master sample library of the MCB dataset provided by the original paper was also used. Peptides that belong to multiple proteins were discarded. For the second species, we extracted *Arabidopsis* precursors from a spectral library built by Zhang et al.[51]. These precursors were filtered to discard sequences that exist in the sample-specific libraries. Next, we spiked them into the sample-specific libraries, where the numbers were exactly the same as the sample-specific species precursors. Top six fragment ions with the highest intensities for each precursor were retained.

Subsequently, all the sample files were processed by the software tools with the resulting two-species libraries. For DreamDIA, centroided mzXML files were used as input. For Skyline and DIA-NN, centroided mzML files were used. For OpenSWATH, profile mzXML files were used as recommended[69]. Each software tool first outputs all the identification results without default FDR control, so that the proxy FDR by two-species library method could be calculated. Results at 1% precursor FDR for each tool were also retained for evaluation (Fig. 3d–f and Supplementary Data 3).

**Evaluation of the contributions for each XIC type in RSM**. We evaluated the contributions for each type of XICs in RSM by stepwise exclusion of the six XIC parts and the *library* part at the other two resolutions. RSMs with eight different compositions containing 20, 40, 60, 110, 120, 130, 150, and 170 XICs, respectively, were tested. First, eight deep representation models were built with the same architecture and hyperparameters except for the input dimension. Then the models were trained and validated on the same training set mentioned above. The best epoch numbers were selected when the validation loss stopped decreasing after at least 10 epochs. Finally, the trained models were used to analyze the S1-1 run in the MCB dataset with both the DIA-Umpire library and DDA master sample library. The numbers of mouse precursors identified at different FDRs with two-species library method were compared.

**Benchmarking of DreamDIA with Avant-garde**. We compared the identification and quantification performance of DreamDIA with Skyline and Avant-garde on the LFQbench HYE110 dataset. We followed the evaluation methods as in the Avant-grade paper[31], while the analysis results of Skyline and Avant-garde were directly obtained from its supplementary materials. For DreamDIA, default settings were used and results at 1% precursor FDR were retained. The number of identifications and valid ratios were obtained from the R objects generated by LFQbench software suite.

**Evaluation of deamidated peptide identification performance**. We followed the method proposed in DIA-NN github repository (https://github.com/vdemichev/DiaNN#ptms) to evaluate the identification confidence of deamidated peptides. The PCS software should identify more deamidated peptides when the correct deamidation mass shift 0.9840 Da is given, while as few identifications as possible when a close pseudo mass shift 1.0227 Da is given.

We first built the sample-specific library using DIA-Umpire as follows. The raw data files were transformed into centroided mzXML files by ProteoWizard[59] (v3.0.19317) with all the other arguments unchecked. The resulting mzXML files were processed by DIA-Umpire (v2.0) to generate pseudo MS/MS spectra. X! Tandem[60] and Comet[61] were used to search these pseudo spectra with deamidation modifications added in the searching parameters, and the searching results were further filtered by PeptideProphet[62] and ProteinProphet[63] in TPP[64,65] (v5.2.0). The sample-specific spectral libraries were subsequently generated by SpectraST[66].

Then the mass-to-charge ratios of deamidation-related ions were added with $N \times (1.0227 - 0.9840)/C$ to obtain the pseudo-modification library, where $N$ denotes the number of deamidation modifications and $C$ denotes the ion charge. Subsequently, equivalent *Arabidopsis* precursors were spiked into the sample-specific library and the pseudo-modification library respectively. 

Finally, the S1-1 run on the MCB dataset was analyzed twice using these two libraries respectively for both DreamDIA and DIA-NN and results without default FDR control were used for comparison. The mass shift of deamidation in DreamDIA was also modified to 1.0227 Da while analyzing the pseudo-modification library.

**Statistics and reproducibility**. No statistical tests were involved in this study. In addition, all the data used are public datasets and all the codes used are publicly available at Github to guarantee the reproducibility of all the experiments.

**Reporting summary**. Further information on research design is available in the Nature Research Reporting Summary linked to this article.

## Data availability

The datasets used in this study are publicly available at the ProteomeXchange Consortium via the PRIDE[70] or iProX[71] partner repository. The dataset identifiers are PXD015098, PXD021390, PXD011691, PXD005573, PXD016647 and PXD002952. Source data for the graphs and charts in the main figures is available in the Supplementary Data files and any remaining information can be obtained from the corresponding author upon reasonable request.

## Code availability

DreamDIA is open-source and available at https://github.com/xmuyulab/DreamDIA-XMBD.

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

## Acknowledgements

This work was supported by the National Natural Science Foundation of China (81788101 to J.H.). We thank Dr. Zhuobin Xu and Dr. Yuwei Yu for help in using the high-performance computer. We thank Shun Wang for providing help for the docker image.

## Author contributions

M.G. and R.Y. designed the study and the algorithms. M.G., W.Y., and R.Y. wrote the first manuscript. M.G. and W.Y. implemented the algorithms. M.G., C.L., Y.C., Y.L., Q.H., and C.-Q.Z. performed the experiments. J.S. and J.H. advised on the algorithm and experiment designs. All authors discussed and commented on the manuscript.

## Competing interests

R.Y. and W.Y. are shareholders of Aginome Scientific. The authors have no further financial or nonfinancial competing interests.
