## [Transparent Peer Review File · Communications Biology]

Reviewers' comments:

Reviewer #1 (Remarks to the Author):

The manuscript by Gao et al reports DreamDIA-XMBD, a deep learning-based tool that can extract and score chromatogram features, improving the performance of peptide-centric DIA data analysis. In contrast to the existing tools that only consider several transitions, DreamDIA-XMBD utilizes additional elution profiles of all the theoretical fragment ions including isotopic peaks by a deep representation model. This feature provides interesting insight into DIA data analysis and warrants in-depth investigation. However, there are some major concerns and remarks should be addressed before publication.

1. The authors should describe the two-organism test in the main text before interpreting the results. Otherwise, it is hard to understand what are "control precursors" and "under the same control levels". Does the term "target" stand for organism-specific (human) precursors? Otherwise, were human and "control" (non-human) precursors combined and used as "targets", and were "decoys" generated on the whole target library including "controls"?

2. The term "control" is confusing. On page 8, the authors state "the decoy precursors were taken as confirmed negative CONTROL, and the target precursors were treated as unlabeled class as they may also include negative precursors". In Supplementary Figure S5, the label "Control" stands for precursors from other organisms not present in the sample. I suggest use another word instead of "control" in the two-organism test.

3. Please report whether the FDR estimated by two-organism test is comparable to decoy-based FDR set in DreamDIA-XMBD. A conservative FDR estimate can be obtained according to the DIA-NN paper (doi:10.1038/s41592-019-0638-x). In a real scenario, user will not perform two-organism test, and can only set an FDR cutoff (e.g., 1%) when using DreamDIA-XMBD.

4. Please provide more details of the representation on page 12:
Are the 50 top XICs in the self part belong to the same fragments and ranked in the same order across all the runs?
How many isotopic peaks are considered for each precursor? M+1 to M+10 for both MS1 and unfragmented quality-tier 3? Are they in the same order across all the runs?
For doubly-charged fragment ions, is 0.5 Da or 1 Da used as the m/z differences of (hypothetical) isotopic peaks?
What is the RT width of 12 cycles in seconds or minutes?

5. Besides precursors, please also report the numbers of identifications, as well as overlap of identifications by DreamDIA-XMBD and other tools, at peptide and protein level.

6. Is the top six fragments associated with each precursor used in quantification (on page 15) ranked by peak area or weighted peak area?

7. DIA-Umpire, PECAN and directDIA should not be classified as "in silico prediction algorithms" (on page 5).

Reviewer #2 (Remarks to the Author):

Gao et al. report a data-driven approach for the analysis of data-independent acquisition (DIA) data, termed DreamDIA-XMBD. The main point in this manuscript is the representation of chromatographic features (XICs) in an LSTM neural network. The authors benchmark their new software against state-

of-the-art tools, demonstrating higher numbers of precursor identifications and precise quantification in public test data sets. To strengthen the conclusions, the authors should try to disentangle the causes of higher identification numbers in more detail, and in particular the effect the higher number of ions per precursor in their approach. Overall, I believe the manuscript is a good fit for the journal upon addressing the points detailed below.

1) Please explain the abbreviation "DreamDIA-XMBD".

2) I understand that the main reason for the improved performance is considering XICs of many more ion species associated with a given peptide precursor increases sensitivity. Hence, to support this claim, it would be interesting to compare

a. The performance of DreamDIA-XMDB with a 'regular' no. of ions/precursor.

b. The performance of another tool such as DIA-NN with a greatly extended no. of ions/precursor.

3) A lot of information is somewhat hidden in the Methods section and the authors should consider adding more detail to the main text. For example, from the main text it is not clear how the additional ions are calculated and based on what. However, this is one of the key points of the present work.

4) The author's approach appears a bit counter-intuitive as one might think that a preferred use of deep learning for DIA is to predict the most likely top fragment ions rather than using all theoretically possible ions to reduce noise. Could the authors comment on this in their manuscript?

5) The choice of the FDR model can greatly influence the number of identified precursors as the authors acknowledge. I would thus appreciate a comparison of the different tools using the same decoy database.

6) I appreciate the two-species test, yet wouldn't it be more appropriate to use a similarly sized proteome to the human proteome such as Arabidopsis as a true false population? My concern is that the current approach is prone to underestimating the FDR.

7) Please clarify in the main text whether the FDR refers to precursor level or protein level, or both. This is also important for a fair comparison between tools.

8) Fig. 1 c: can the authors clarify their definition of an 'in silico' library? Can DreamDIA-XMBD perform this analysis independent of other tools such as DIA-Umpire?

9) While different numbers are presented for different software tools, I miss a more in-depth analysis of the DreamDIA-unique precursors. For example, are the latter typically low-abundance precursors? I would suggest to include such analysis as a main figure.

10) Fig 2: I think this analysis contains only limited information as it is limited to the most abundant proteins. The authors should consider moving this figure to the Supplementary Information. Out of curiosity: the number of precursors/protein appears very high. How do these numbers translate to peptides and protein coverage?

11) I was surprised to read that only the top six fragment ions were used for quantification (just as with standard approaches). Wouldn't it follow the reasoning of the authors to use as many ions as possible as they do for identification?

12) I would appreciate a more in-depth comparison to another 'data-driven' DIA analysis approach: 'Avant-Garde' by Jaffe and colleagues (ref. 31).

13) Could this approach be extended to ion mobility-based DIA workflows such as diaPASEF?

14) Do the authors know whether deamidation could be a problem with this approach?

15) Suppl. Fig. S4: Please start the y-axis at 0.

16) Please comment on the time needed for computation and system requirements.

Reviewer #3 (Remarks to the Author):

Summary

The authors describe a novel algorithm for the identification (and quantification) of peptide precursors from DIA data. Instead of using hand engineered scores extracted from DIA data they use a deep neural network (LSTM) applied to a matrix of extracted ion currents (XICs). The 16-dimensional deep representation features of the neural network, together with some more classical scores are fed into various classifier types to deliver a final discriminant score. They authors show a roughly 7% increase in precursor identifications compared to state of the art. This work is relevant and reflects a trend towards end to end learning applied to DIA data analysis. Many important aspects of the work are only vaguely described and investigated. The manuscript could improve in clarity. Given the relatively small improvements in precursor identifications more time should be spent in evaluating various versions of models and input data (XICs). I recommend to publish this work with major revisions.

Major Points

- page 2, next to Ting et al. please also cite Gillet et al. 2012 as this has been the first publication to apply peptide-centric analysis to DIA data. Ting et al. introduced the terminology.
- page 3, second line: please cite Rosenberger et al. 2017 (Nature Methods) and Reiter et al. 2011 (Nature Methods) as these papers describe the methods used nowadays for FDR control in peptide-centric analysis.
- page 4, line 2: please also cite Bruderer et al. 2015 (MCP). Spectronaut is one of the more widely used PCS software packages.
- page 4, Please describe in more detail how training with positive-unlabeled (PU) data points was performed.
- page 6, paragraph 2: Please describe what exactly is meant with "in silico" library? In figure 1c there is only one line for DreamDIA-XMBD. Based on the text I would have expected 2 lines.
- page 7 or figure 2: The test with the high confident proteins (HCPs) is not very convincing. Any algorithm that would simply identify more (wrong) precursors would show an increase in precursors identified for HCPs. I would recommend to remove this figure completely.
- page 11: "The library part contains 20 top extracted ion chromatograms (XICs) at three different resolutions of fragment ions in the spectral library". Please describe this better. 20 is cannot be divided by 3? In general the various XICs are very poorly described.
- page 16: please describe more explicitly how FDR was estimated using the two species test.
- Figure 4: Please benchmark the new model (precursor ID improvements @1% FDR) on more data sets. I'd recommend to condense this information into one figure.
- Figure 5: I'd recommend to move this figure to the supplementary as this paper is primarily about identification and not quantification.
- Please describe whether you envision this model to be trained once and then applied to data sets or whether you see it realistic that this model is trained on the fly on every new data set. Please describe how you envision this model to be used. How long does it take to train the model?
- The authors should show the effect of including/excluding various XIC types (library, self, qt3, ms1, iso, and light) into their model. Please also describe significant hyperparameter search performed on the model.
- Supplementary Figure S2 is very anecdotic. Please make a more systematic analysis on the additionally identified precursors, e.g. on the additionally identified precursors compared to DIA-NN. Do these precursors have MS1 features? If not there is a risk that the additional precursors are in fact

modified peptides that are mistakingly identified as their non-modified counterparts.

Minor Points

- page 2, last paragraph: I'd prefer a different term over "...noisy spectra...", e.g. "...convoluted spectra originated from signals...". The signals in these spectra are from real peptides and largely used in DIA analysis.
- page 2, last line: The term "transition" derives from the MRM world. In the context of DIA I'd prefer the term fragment ion.
- page 4, this sentence sounds a bit odd: "The deep representation model in DreamDIA-XMBD demonstrates its capability to extract features from the complex elution patterns in RSM, which may bring interference in conventional heuristic peptide scoring systems."
- page 5, second to last line: please be more clear with what you mean with "...under the same control levels..."
- Please state how many epochs the model was trained and using what kind of hardware
- Figure 3: Please introduce dds also in the text

Response to Reviewers' Comments

Response to Reviewers

Reviewer: 1

Comments:

The manuscript by Gao et al reports DreamDIA-XMBD, a deep learning-based tool that can extract and score chromatogram features, improving the performance of peptide-centric DIA data analysis. In contrast to the existing tools that only consider several transitions, DreamDIA-XMBD utilizes additional elution profiles of all the theoretical fragment ions including isotopic peaks by a deep representation model. This feature provides interesting insight into DIA data analysis and warrants in-depth investigation. However, there are some major concerns and remarks should be addressed before publication.

Authors' response: We thank the reviewer for careful reading and the valuable comments. We have carefully considered the comments and have revised the manuscript accordingly.

1. The authors should describe the two-organism test in the main text before interpreting the results. Otherwise, it is hard to understand what are "control precursors" and "under the same control levels". Does the term "target" stand for organism-specific (human) precursors? Otherwise, were human and "control" (non-human) precursors combined and used as "targets", and were "decoys" generated on the whole target library including "controls"?

Authors' response: In the revised manuscript, we have described in more details about the two-species library method in the first paragraph of the *Results* section as follows.

"As different software tools use different strategies for FDR estimation, it is difficult to compare the identification performance directly. Herein, we adopted the two-species spectral library method for benchmarking, where the same number of Arabidopsis precursors were added to the mouse sample-specific spectral libraries as false positive controls. Effective FDR at precursor level was calculated as the number of Arabidopsis precursors identified divided by the number of all precursors identified."

Moreover, we have used a more intuitive figure, which had also been used in the DIA-NN paper, to show the results (Figure 2).

2. The term "control" is confusing. On page 8, the authors state "the decoy precursors were taken as confirmed negative CONTROL, and the target precursors were treated as unlabeled class as they may also include negative precursors". In Supplementary Figure S5, the label "Control" stands for precursors from other organisms not present in the sample. I suggest use another word instead of "control" in the two-organism test.

Authors' response: We thank the reviewer for this suggestion. In the revised manuscript, the “target” and “control” have been replaced by the direct species names, namely, “mouse/human” (true positive precursors) and “Arabidopsis” (false positive precursors).

3. Please report whether the FDR estimated by two-organism test is comparable to decoy-based FDR set in DreamDIA-XMBD. A conservative FDR estimate can be obtained according to the DIA-NN paper (doi:10.1038/s41592-019-0638-x). In a real scenario, user will not perform two-organism test, and can only set an FDR cutoff (e.g., 1%) when using DreamDIA-XMBD.

Authors' response: We have performed identification evaluation with both two-species FDR method and decoy-based FDR method, and have compared the identification consistency with the other state-of-the-art software tools (Figure 3) in the revised manuscript. DreamDIA shows high identification consistency with other software tools when using both FDR methods, which indicates that DreamDIA can produce good identification results with conservative FDR estimation.

4. Please provide more details of the representation on page 12:

Are the 50 top XICs in the self part belong to the same fragments and ranked in the same order across all the runs?

How many isotopic peaks are considered for each precursor? $M+1$ to $M+10$ for both MS1 and unfragmented quality-tier 3? Are they in the same order across all the runs?

For doubly-charged fragment ions, is 0.5 Da or 1 Da used as the m/z differences of (hypothetical) isotopic peaks?

What is the RT width of 12 cycles in seconds or minutes?

Authors' response: We have added a new subsection, “Impact of representative spectral matrix design to identification performance”, in Results to discuss more about the RSM design in the revised manuscript. We also described the XIC compositions of the RSM in details in the “Deep representation model in DreamDIA” subsection of Methods, which gives a detailed description of the representation model as follows.

“The input to the deep representation model is the RSM (Figure 1b), a matrix consisting of 170 XICs across six types of elution profiles, namely, library, self, qt3, ms1, iso and light. The library part contains 60 XICs in total, which include fragment ions in the spectral library at three different resolutions, r , $0.2 * r$ and $0.45 * r$, where r denotes the basic resolution in ppm or Da. For each resolution, 20 XICs with highest intensities are kept. Zero-filling is used if less than 20 fragments are available in the library. After extraction, the XICs at the basic resolution are sorted by the sum of their Pearson correlations with all the other library XICs. Once the order of XICs at the basic resolution is fixed, XICs at the other two resolutions are sorted according to their corresponding relationships to the XICs at the basic resolution. The iso part contains 20 XICs at $(M+1)/q$ of each library fragment, which correspond to potential isotopic peaks of these fragments. In addition, XIC at $(M-1)/q$ of each library fragment, which indicates the possibility of this fragment being actually a heavy isotopologue of a light fragment, is also included as the light part of RSM. The self part contains XICs of all the theoretical fragment ions from one precursor. Specifically, for precursors

with two charges, fragment ions with one charge are considered. For precursors with charges greater than two, fragment ions with one and two charge(s) are considered. Intensity-based filtering or zero filling is used when more or less than 50 fragments are available. After extraction, the *self* XICs are sorted by their Pearson correlations with the first *library* XIC. Besides fragment ions, we also include XICs of precursor ion at the three aforementioned resolutions, its (M+1)/q to (M+4)/q isotopic peaks and the XIC at (M-1)/q, both at the basic resolution into the *ms1* part. XICs of unfragmented precursor ion and its (M+1)/q to (M+4)/q isotopic peaks are also considered as the *qt3* part. The XICs in these two parts are in a fixed order for all RSMs. The RT width of the RSM is set to 12 cycles (around 3 seconds per cycle for most equipments) by default, which is long enough for most elution signals as we manually verified on several DIA datasets.”

5. Besides precursors, please also report the numbers of identifications, as well as overlap of identifications by DreamDIA-XMBD and other tools, at peptide and protein level.

Authors’ response: We thank the reviewer for this suggestion. We have reported the consistency of precursors, peptides and proteins identified by DreamDIA compared with the other software tools using both the two-species FDR method and the decoy-based FDR method in the revised manuscript (Figure 3).

6. Is the top six fragments associated with each precursor used in quantification (on page 15) ranked by peak area or weighted peak area?

Authors’ response: The top six fragments for quantification were *library* fragments ranked by the sum of their Pearson correlations with all the other *library* XICs. In the revised manuscript, we have added the following sentence in the *Methods* section to clarify this point.

“In general, the top six *library* fragments associated with each precursor sorted by the sum of their Pearson correlations with all the other *library* XICs were used for quantification.”

7. DIA-Umpire, PECAN and directDIA should not be classified as “*in silico* prediction algorithms” (on page 5).

Authors’ response: Thanks for the comments. As appropriate, we now used “spectra deconvolution algorithms” to describe DIA-Umpire and directDIA in the revised manuscript as follows, and PECAN has been discarded.

“Typically, as for routine analysis of DIA-MS data, the spectral libraries can be built from DDA master samples containing the same proteins with the corresponding DIA data, or from the DIA data directly with the aid of spectra deconvolution algorithms such as DIA-Umpire and directDIA in Spectronaut. In this study, we evaluated the identification performance with spectral libraries from both sources.”

Reviewer: 2

Comments:

Gao et al. report a data-driven approach for the analysis of data-independent acquisition (DIA) data, termed DreamDIA-XMBD. The main point in this manuscript is the representation of chromatographic features (XICs) in an LSTM neural network. The authors benchmark their new software against state-of-the-art tools, demonstrating higher numbers of precursor identifications and precise quantification in public test data sets. To strengthen the conclusions, the authors should try to disentangle the causes of higher identification numbers in more detail, and in particular the effect the higher number of ions per precursor in their approach. Overall, I believe the manuscript is a good fit for the journal upon addressing the points detailed below.

Authors' response: We thank the reviewer for careful reading and the valuable comments. We have carefully considered the comments and have revised the manuscript accordingly.

1. Please explain the abbreviation "DreamDIA-XMBD".

Authors' response: "Dream" means Deep REpresentAtion Model, we have added the explanation in the *Abstract*. "XMBD" means Xiamen Big Data, which refers to the biomedical open software initiative in the National Institute for Data Science in Health and Medicine, Xiamen University, China. We have explained "XMBD" with a footnote on its first appearance in the *Abstract* and denoted the proposed software as "DreamDIA" in the rest of the revised manuscript.

2. I understand that the main reason for the improved performance is considering XICs of many more ion species associated with a given peptide precursor increases sensitivity. Hence, to support this claim, it would be interesting to compare

a. The performance of DreamDIA-XMDB with a 'regular' no. of ions/precursor.

b. The performance of another tool such as DIA-NN with a greatly extended no. of ions/precursor.

Authors' response: It is true that the inclusion of more ion species plays an important role in peptide identification of DreamDIA. To clarify the contributions of each XIC species in the RSM to the identification results, we have performed more experiments and added a new subsection, "*Impact of representative spectral matrix design to identification performance*", in the revised manuscript.

First, we trained eight deep representation models with different XIC compositions as input, and evaluated their peptide identification performance (Figure 4). The first model whose input RSM contained 20 XICs of fragments in the spectral library was actually equal to the "regular" method adopted by all the conventional software tools. The results confirmed that removing any XIC part from RSM led to performance drop, which confirmed the contribution of all the XIC parts included in our design.

Second, we used the SHAP deep explainer to show the feature importance distributions in the RSM (Supplementary Figure S5), which provides a more direct illustration on the contributions of different XIC parts to the final identification results.

Although it would be interesting to study if other algorithms such as OpenSWATH and DIA-NN could benefit from including more fragment ions also, it is actually very difficult to conduct such an experiment as these tools use highly complex and sophisticated manually engineered feature calculation rules to process XICs. As a result, these software tools do not support including these additional XIC types unless their discriminative models are redesigned.

3. A lot of information is somewhat hidden in the Methods section and the authors should consider adding more detail to the main text. For example, from the main text it is not clear how the additional ions are calculated and based on what. However, this is one of the key points of the present work.

Authors' response: We thank the reviewer for the advice. We have added a new subsection, “*Impact of representative spectral matrix design to identification performance*”, in the *Results* section to discuss more about the RSM design in the revised manuscript.

4. The author's approach appears a bit counter-intuitive as one might think that a preferred use of deep learning for DIA is to predict the most likely top fragment ions rather than using all theoretically possible ions to reduce noise. Could the authors comment on this in their manuscript?

Authors' response: This is indeed an important point behind the design philosophy of DreamDIA. In fact, DreamDIA, as in other deep learning-based approaches for tasks such as image or speech recognition, adopts an “end-to-end” design philosophy where the model is supplied with all the relevant data and let the deep learning algorithm to figure out the most relevant ones (or their combination) for prediction.

In the revised manuscript, we have elaborated this idea in the *Discussion* section as follows.

“Of note, one important characteristic of data-driven approaches is that they can solve complex machine learning problems by simple statistical models supplied with all relevant data available rather than depending on customized features derived from domain knowledge. However, it can be anticipated that the performance of such a model will be highly associated with the training data size. Indeed, in many data-driven machine learning tasks such as object recognition and natural language processing, there is a clear association of increased benchmark performance with increasing amount of training data. In this perspective, it is anticipated that the performance of DreamDIA could further improve with the availability of more high-quality training data in near future.”

5. The choice of the FDR model can greatly influence the number of identified precursors as the authors acknowledge. I would thus appreciate a comparison of the different tools using the same decoy database.

Authors' response: Thanks a lot for the comment. The FDR model indeed makes a difference in the number of identifications. Therefore, we adopted the two-species library method as used in other literature to calculate effective FDR for fair comparison so that for most experiments, the decoy database used for FDR calculation was actually the spiked-in Arabidopsis precursors, which was completely identical for all the software tools. The effective FDR was calculated as the number of Arabidopsis (false positive) precursors identified divided by the number of all precursors identified for each software tool.

On the other hand, when the two-species library is not used for effective FDR estimation, it is a bit challenging for other existing software tools to use the same decoy database due to the compatibility of their discriminative models with the decoy generation methods. To overcome this limitation, in the updated version of DreamDIA, we have implemented five commonly-used decoy generation methods (Supplementary Figure S6a) in our software, and performed a comprehensive evaluation of all the compatible decoy generation methods for DreamDIA, DIA-NN, OpenSWATH and Skyline (Supplementary Figure S6b). Almost all the decoy generation methods present high stability for peptide identification performance, except the random mass shift algorithm used by Skyline that induced performance drop. This result reflects not only the stability for all the software tools even when the decoy databases are generated by different methods, but also the reliability of our software when used without two-species library (Figure 3b, c, d and all the LFQbench tests).

6. *I appreciate the two-species test, yet wouldn't it be more appropriate to use a similarly sized proteome to the human proteome such as Arabidopsis as a true false population? My concern is that the current approach is prone to underestimating the FDR.*

Authors' response: Thanks for the comment. It is indeed a very good suggestion. In the revised manuscript, we used an Arabidopsis spectral library in place of the original yeast and E.coli library as true false population. For each mouse or human sample-specific library in this work, exactly the same number of Arabidopsis precursors were added as false positive targets to build the two-species library.

7. *Please clarify in the main text whether the FDR refers to precursor level or protein level, or both. This is also important for a fair comparison between tools.*

Authors' response: We have clarified the FDR levels throughout the main text of the revised manuscript. An example of the modified sentences in the *Results* section is as follows.

“Effective FDR at precursor level was calculated as the number of Arabidopsis precursors identified divided by the number of all precursors identified.”

8. *Fig. 1 c: can the authors clarify their definition of an ‘in silico’ library? Can DreamDIA-XMBD perform this analysis independent of other tools such as DIA-Umpire?*

Authors' response: We agree that the term ‘in silico’ library is ambiguous. We have replaced this

description with “DIA-Umpire library” directly, and clarified its difference between the DDA master sample library in the revised manuscript.

“Typically, as for routine analysis of DIA-MS data, the spectral libraries can be built from DDA master samples containing the same proteins with the corresponding DIA data, or from the DIA data directly with the aid of spectra deconvolution algorithms such as DIA-Umpire and directDIA in Spectronaut.”

In addition, DreamDIA also supports other spectral libraries and we have also performed evaluations when using libraries from both DIA-Umpire and DDA master sample (Figure 1, Supplementary Figure S1 and Figure 4).

9. While different numbers are presented for different software tools, I miss a more in-depth analysis of the DreamDIA-unique precursors. For example, are the latter typically low-abundance precursors? I would suggest to include such analysis as a main figure.

Authors’ response: In the revised manuscript, we have performed more comprehensive analysis of the identification results, such as consistency of the identification results with the other software tools under both two-species FDR method and decoy-based default FDR method and intensity distribution evaluation (Figure 3). And we also evaluated the abundance distribution of DreamDIA-unique precursors (Figure 3h). Results show that the abundances of these precursors have a Gaussian-like distribution, which indicates that DreamDIA does not have abundance bias for peptide identification. In addition, several elution profile examples of DreamDIA-unique precursors were shown in Supplementary Figure S2.

10. Fig 2: I think this analysis contains only limited information as it is limited to the most abundant proteins. The authors should consider moving this figure to the Supplementary Information. Out of curiosity: the number of precursors/protein appears very high. How do these numbers translate to peptides and protein coverage?

Authors’ response: We thank the reviewer for pointing out this issue. As suggested by other reviewers as well, we have replaced this figure with a more comprehensive analysis of the identification results in the revised manuscript (Figure 3).

11. I was surprised to read that only the top six fragment ions were used for quantification (just as with standard approaches). Wouldn’t it follow the reasoning of the authors to use as many ions as possible as they do for identification?

Authors’ response: We thank the reviewer for this good question. Note that the deep representation model in DreamDIA has the ability to denoise and extract useful information from the RSMs for precursor identification, while there is still complex interference in these theoretical XICs. Therefore, inclusion of more ions in quantification stage should be more careful for the PCS software. To validate this concept, we have also conducted experiments to include more fragment ions for quantification and the results are in consistence with this notion. We have included these

new results in the revised manuscript as follows.

“We further tested whether the inclusion of more fragments would affect the quantification accuracy. We modified the original quantification calculation to include top 3 to 15 *self* fragments sorted by their Pearson correlations with the first *library* fragment, and their Pearson correlation weighted areas were added to the final results. Inclusion of these fragments has relatively small impact on the quantification results, with slight improvement on yeast peptide and protein quantification but slight degradation on E.coli peptide and protein quantification in terms of deviation of the measured ratios to the groundtruth (Supplementary Figure S11, S12 and Supplementary Table S4). It is noteworthy that although the deep representation model can extract useful information from the RSMs for precursor identification, many theoretical XICs may contain complex interference in their elution profile waveforms. Therefore, inclusion of XICs with poor signal-to-noise ratio without performing sophisticated denoising may not improve quantification results.”

12. *I would appreciate a more in-depth comparison to another ‘data-driven’ DIA analysis approach: ‘Avant-Garde’ by Jaffe and colleagues (ref. 31).*

Authors’ response: We have benchmarked DreamDIA with Avant-Garde in a new subsection, “*Benchmark of DreamDIA with Avant-garde*”, in the revised manuscript. As expected, DreamDIA could identify more peptides than both Skyline and Avant-garde (Supplementary Figure S13a). For quantification, DreamDIA also achieved higher accuracy as indicated by its closer medians of the calculated log-ratios to the expected values compared with Skyline and Avant-garde, while Avant-garde showed lower quantification variation than DreamDIA due to its relatively aggressive fragment filtering operation (Supplementary Figure S13b, c, d).

13. *Could this approach be extended to ion mobility-based DIA workflows such as diaPASEF?*

Authors’ response: It is indeed a very good question. DiaPASEF is recently one of the most cutting-edge technologies in DIA research field which we have also paid attention to. For the data structure of diaPASEF, there are the same three elements, *m/z*, RT and intensity compared with routine DIA data, as well as one more dimension of ion mobility. Therefore, in principle, this type of data structure is still proper for deep representation models to extract relevant features. As the increasing of the amount of public diaPASEF datasets, training a deep representation model for diaPASEF spectrum can be also feasible. Considering plenty of efforts have to be made including data collection, spectra preprocessing, model design and optimization, model training and validation and so forth, we will keep working on the supporting of diaPASEF data in DreamDIA in our future work.

14. *Do the authors know whether deamidation could be a problem with this approach?*

Authors’ response: We have evaluated DreamDIA for the identification of deamidated peptides with the method proposed by the authors of DIA-NN ([#ptms](https://github.com/vdemichev/DiaNN)) and the results are provided in the revised manuscript (Supplementary Figure S14). Results show that DreamDIA can confidently identify more deamidated peptides compared with DIA-NN.

15. *Suppl. Fig. S4: Please start the y-axis at 0.*

Authors' response: We thank the reviewer for careful reading. We have removed this figure due to its redundant information with the new Figure2 and FigureS1.

16. *Please comment on the time needed for computation and system requirements.*

Authors' response: We have commented on the time needed for computation and system requirements in a new subsection, “*Computational performance*”, in the revised manuscript.

Reviewer: 3

Comments:

The authors describe a novel algorithm for the identification (and quantification) of peptide precursors from DIA data. Instead of using hand engineered scores extracted from DIA data they use a deep neural network (LSTM) applied to a matrix of extracted ion currents (XICs). The 16-dimensional deep representation features of the neural network, together with some more classical scores are fed into various classifier types to deliver a final discriminant score. They authors show a roughly 7% increase in precursor identifications compared to state of the art. This work is relevant and reflects a trend towards end to end learning applied to DIA data analysis. Many important aspects of the work are only vaguely described and investigated. The manuscript could improve in clarity. Given the relatively small improvements in precursor identifications more time should be spent in evaluating various versions of models and input data (XICs). I recommend to publish this work with major revisions.

Authors' response: We thank the reviewer for careful reading and the valuable comments. We have carefully considered the comments and have revised the manuscript accordingly.

1. *page 2, next to Ting et al. please also cite Gillet et al. 2012 as this has been the first publication to apply peptide-centric analysis to DIA data. Ting et al. introduced the terminology.*

2. *page 3, second line: please cite Rosenberger et al. 2017 (Nature Methods) and Reiter et al. 2011 (Nature Methods) as these papers describe the methods used nowadays for FDR control in peptide-centric analysis.*

3. *page 4, line 2: please also cite Bruderer et al. 2015 (MCP). Spectronaut is one of the more widely used PCS software packages.*

Authors' response: We thank the reviewer for careful reading. We have added these important citations in the revised manuscript.

4. *page 4, Please describe in more detail how training with positive-unlabeled (PU) data points was performed.*

Authors' response: Thanks for the comments. In the revised manuscript, we have added more details about the training process of the discriminant models in the *Methods* section as follows:

“Due to the existence of false targets in a spectral library, this model is trained based on the principle of PU learning with decoy peptides as affirmative negative controls while treating targets as unlabeled. We adopted the naive approach that training a binary classifier directly between confirmed decoys and undetermined target precursors to estimate the probability for an RSM of belonging to a real peptide. XGBoost was chosen as the default discriminative model for its superior performance (Results) and high efficiency. The software also provides an option for users to choose other types of classifiers such as random forest as the discriminative model. To further reduce the bias of RT dislocation between spectral libraries and DIA runs, we adopt the test-time augmentation strategy where all the RSMs within a certain range centered at the predicted RT with dds higher than a given threshold were included as potential targets for a precursor in the PU learning process. All the features of candidate RSMs are used to train a binary classifier with depths of 6 and 12 for XGBoost and random forest models respectively to assign a discriminant score for each RSM. The number of estimators of random forest model is set to 200 by default.”

5. page 6, paragraph 2: Please describe what exactly is meant with "in silico" library? In figure 1c there is only one line for DreamDIA-XMBD. Based on the text I would have expected 2 lines.

Authors' response: Thanks for the comment. It is indeed very confusing to use the term “in silico” library. In the revised manuscript, we have replaced this description that may confuse readers with “DIA-Umpire library” directly, and clarified its difference between the DDA master sample library as follows:

“Typically, as for routine analysis of DIA-MS data, the spectral libraries can be built from DDA master samples containing the same proteins with the corresponding DIA data, or from the DIA data directly with the aid of spectra deconvolution algorithms such as DIA-Umpire and directDIA in Spectronaut. In this study, we evaluated the identification performance with spectral libraries from both sources.”

Moreover, we have used a more intuitive figure, which had also been used in the DIA-NN paper and other publications, to show the results (Figure 2 of the revised manuscript). The number of mouse precursors identified at different effective FDRs by the four software tools were compared.

6. page 7 or figure 2: The test with the high confident proteins (HCPs) is not very convincing. Any algorithm that would simply identify more (wrong) precursors would show an increase in precursors identified for HCPs. I would recommend to remove this figure completely.

Authors' response: We thank the reviewer for pointing out this issue. We have removed this figure in the revised manuscript. Instead, we performed more comprehensive analysis of the identification results in the revised manuscript including consistency with the other software tools using both two-

species FDR method and decoy-based default FDR method and intensity distribution evaluation (Figure 3 of the revised manuscript).

7. page 11: "The library part contains 20 top extracted ion chromatograms (XICs) at three different resolutions of fragment ions in the spectral library". Please describe this better. 20 is cannot be divided by 3? In general the various XICs are very poorly described.

Authors' response: We have added a new subsection, "Impact of representative spectral matrix design to identification performance", in the *Results* section to discuss more about the RSM design in the revised manuscript. Furthermore, we have also described the XIC compositions of the RSM in detail in the "Deep representation model in DreamDIA" subsection of the *Methods* section as follows.

"The input to the deep representation model is the RSM (Figure 1b), a matrix consisting of 170 XICs across six types of elution profiles, namely, library, self, qt3, ms1, iso and light. The library part contains 60 XICs in total, which include fragment ions in the spectral library at three different resolutions, r , $0.2 * r$ and $0.45 * r$, where r denotes the basic resolution in ppm or Da. For each resolution, 20 XICs with highest intensities are kept. Zero-filling is used if less than 20 fragments are available in the library. After extraction, the XICs at the basic resolution are sorted by the sum of their Pearson correlations with all the other library XICs. Once the order of XICs at the basic resolution is fixed, XICs at the other two resolutions are sorted according to their corresponding relationships to the XICs at the basic resolution. The iso part contains 20 XICs at $(M+1)/q$ of each library fragment, which correspond to potential isotopic peaks of these fragments. In addition, XIC at $(M-1)/q$ of each library fragment, which indicates the possibility of this fragment being actually a heavy isotopologue of a light fragment, is also included as the light part of RSM. The self part contains XICs of all the theoretical fragment ions from one precursor. Specifically, for precursors with two charges, fragment ions with one charge are considered. For precursors with charges greater than two, fragment ions with one and two charge(s) are considered. Intensity-based filtering or zero filling is used when more or less than 50 fragments are available. After extraction, the self XICs are sorted by their Pearson correlations with the first library XIC. Besides fragment ions, we also include XICs of precursor ion at the three aforementioned resolutions, its $(M+1)/q$ to $(M+4)/q$ isotopic peaks and the XIC at $(M-1)/q$, both at the basic resolution into the ms1 part. XICs of unfragmented precursor ion and its $(M+1)/q$ to $(M+4)/q$ isotopic peaks are also considered as the qt3 part. The XICs in these two parts are in a fixed order for all RSMs. The RT width of the RSM is set to 12 cycles (around 3 seconds per cycle for most equipments) by default, which is long enough for most elution signals as we manually verified on several DIA datasets."

8. page 16: please describe more explicitly how FDR was estimated using the two species test.

Authors' response: In the revised manuscript, we have described more about the two-species library method in the first paragraph of the *Results* section as follows.

“As different software tools use different strategies for FDR estimation, it is difficult to compare the identification performance directly. Herein, we adopted the two-species spectral library method for benchmarking, where the same number of Arabidopsis precursors were added to the mouse sample-specific spectral libraries as false positive controls. Effective FDR at precursor level was calculated as the number of Arabidopsis precursors identified divided by the number of all precursors identified.”

Moreover, we have used a more intuitive figure, which had also been adopted in the DIA-NN paper, to show the results more clearly (Figure 2).

9. Figure 4: Please benchmark the new model (precursor ID improvements @1% FDR) on more data sets. I'd recommend to condense this information into one figure.

Authors' response: We have benchmarked DreamDIA using one more dataset for quantification, the LFQbench HYE110 dataset in the revised manuscript. Consider of the size limitation, we have shown the results in Supplementary Figure S9 and S10 respectively for these two datasets. Moreover, the quantification accuracy metric provided by LFQbench software suite has also been shown in Supplementary Table S2 and S3.

10. Figure 5: I'd recommend to move this figure to the supplementary as this paper is primarily about identification and not quantification.

Authors' response: We thank the reviewer for this suggestion. We have moved the quantification benchmarking results to Supplementary Figure S9 and S10 in the revised manuscript.

11. Please describe whether you envision this model to be trained once and then applied to data sets or whether you see it realistic that this model is trained on the fly on every new data set. Please describe how you envision this model to be used. How long does it take to train the model?

Authors' response: In this study, we trained a generic deep representation model with data from three types of commonly-used MS equipment (TripleTOF 5600, Orbitrap Fusion Lumos and Q Exactive HF-X) and obtained considerable improvement on the independent testing datasets (Orbitrap Fusion Lumos, TripleTOF 6600 and Q Exactive HF). Results showed that this trained model has good performance for the data from the MS equipment mentioned above. Thus, we provide this trained model for users to analyze their data without any training procedures. On the other hand, as the development of DIA proteomics, the elution patterns of peptides in DIA data may change over time as the LC-MS technologies evolve. As a consequence, we also provide an advanced API for training customized models for new datasets. The training procedure requires about 3min/epoch for ~1 million RSMs on our GeForce GTX 1080 GPU and the training process typically converges after around 10 epochs. Therefore, it will take only several minutes to train a customized model with a commonly-used GPU if tens of thousands of RSMs are available.

We have also modified our descriptions in the *Introduction* section about how to use the deep representation models as follows.

“DreamDIA provides a deep representation network-based feature extraction method for DIA data analysis, in combination with a novel interface to integrate deep learning algorithms to achieve better performance in large-scale biological and medical proteome research. The training data of the deep representation model can be easily obtained from public datasets. We also provided a trained model that can be directly applied to analyze DIA data with high coverage and accuracy, as well as an application programming interface (API) for customized model. Users can choose our default model for data analysis from widely-used data acquisition equipments conveniently without training, or use our API to train a customized model in minutes that better fits their own experiments.”

12. The authors should show the effect of including/excluding various XIC types (library, self, qt3, ms1, iso, and light) into their model. Please also describe significant hyperparameter search performed on the model.

Authors’ response: We thank the reviewer for this suggestion. We evaluated the effect of excluding various XIC types and the results were shown in Figure 4 in the revised manuscript. Moreover, we used the SHAP deep explainer to visualize the feature importance distributions in the RSM (Supplementary Figure S5). We have also preformed tests on significant hyper-parameters for the deep representation models, which only shows small variance on the peptide identification performance (Supplementary Table S1).

13. Supplementary Figure S2 is very anecdotic. Please make a more systematic analysis on the additionally identified precursors, e.g. on the additionally identified precursors compared to DIA-NN. Do these precursors have MS1 features? If not there is a risk that the additional precursors are in fact modified peptides that are mistakingly identified as their non-modified counterparts.

Authors’ response: Thanks for pointing out this issue. In the revised manuscript, we have added more comprehensive analysis including the identification consistency compared with all the other software tools and the abundance distributions of the identified precursors, peptides and proteins (Figure 3). The results indicate that DreamDIA can produce highly consistent identifications as well as steadily more unique identifications compared with the other software tools. Moreover, we compared the distributions of two MS1-related sub-scores provided by OpenSWATH between the precursors identified by all the four software tools and the ones identified exclusively by DreamDIA (Supplementary Figure S4). The results showed that the majority of those additionally identified precursors by DreamDIA indeed had MS1 signals, and the distributions of these sub-scores were very similar to those of consensus precursors identified by all the software tools.

14. page 2, last paragraph: I'd prefer a different term over "...noisy spectra...", e.g. "...convoluted spectra originated from signals...". The signals in these spectra are from real peptides and largely used in DIA analysis.

Authors’ response: We thank the reviewer for careful reading. We have modified this sentence as follows.

“Despite various advantages of DIA, the challenge of DIA data analysis roots in its convoluted spectra originated from signals of multiple co-fragmented precursor ions.”

15. page 2, last line: *The term "transition" derives from the MRM world. In the context of DIA I'd prefer the term fragment ion.*

Authors' response: We have replaced “transition” with “fragment ion” in both the *Abstract* and the *Introduction* sections in the revised manuscript.

16. page 4, this sentence sounds a bit odd: *"The deep representation model in DreamDIA-XMBD demonstrates its capability to extract features from the complex elution patterns in RSM, which may bring interference in conventional heuristic peptide scoring systems."*

Authors' response: Thanks for pointing out this unclear description. This statement is inaccurate and we have removed this sentence in the revised manuscript.

17. page 5, second to last line: *please be more clear with what you mean with "...under the same control levels..."*

Authors' response: Thanks for the comment. In the revised manuscript, we have adopted a more intuitive approach as used in other literatures to compare the results from the two-species FDR estimation method to avoid the confusion. See Figure 2 and Supplementary Figure S3 in the revised manuscript.

18. *Please state how many epochs the model was trained and using what kind of hardware.*

Authors' response: In the revised manuscript, we have clarified these details in the *Method* section as follows.

“In total, we obtained around 1 million RSMs from the three datasets, which were split by 7:3 as training and validation data for the deep representation model. The validation loss stopped decreasing after 11 epochs. We then retrain the model with all the 1 million RSMs for 11 epochs to obtain the final model. Model training was done on a GeForce GTX 1080 GPU.”

19. *Figure 3: Please introduce dds also in the text.*

Authors' response: In the revised manuscript, we have added the following description on *dds* in the *Results* section to explain this score.

“The deep discriminant score (*dds*), which is calculated by the deep representation model to indicate the probability that each precursor in the spectral library belongs to a real peptide, was also included in our comparison.”

REVIEWERS' COMMENTS:

Reviewer #1 (Remarks to the Author):

My issues with the original manuscript have been well addressed, and I suggest acceptance with an additional minor comment.

I don't think the heading "Identification of PTM peptides with DreamDIA" is fully appropriate since this section only reports the numbers of identified deamidated peptides. Other PTMs are not tested, and the most important task for PTM analysis – site localization – is not involved in this section. The reason that DreamDIA identified more deamidation peptides is not discussed, either.

But this is not the subject of the manuscript, maybe the topic in future investigations, and should not prevent publication of this manuscript. I suggest changing the heading to "Identification of more deamidated peptides with DreamDIA" and move this section to the Supplementary Information.

Reviewer #2 (Remarks to the Author):

The authors have sufficiently addressed my previous comments and greatly improved their manuscript. There are a few grammar mistakes and typos that I am confident will be addressed in the final version.

Two minor points occurred to me:

1) Please define 'RT cycles'. (I think 'acquisition cycle' would be the more common term?)

2) I was wondering whether an experimentally measured mass signal could be (accidentally) assigned to multiple precursors given the large number of transitions considered for each precursor. Could the authors please comment on this?

Reviewer #3 (Remarks to the Author):

The manuscript has improved significantly in clarity and depth of analysis. The authors have addressed all my comments.

There is one final concern to be addressed:

- page 6, "Effective FDR at precursor level was calculated as the number of Arabidopsis precursors identified divided by the number of all precursors identified." This is the FDR estimation procedure as described in literature to be suitable for spectrum-centric analysis (Elias and Gygi Nature Methods 2007). It is not an accurate estimate for FDR in peptide-centric analysis (Reiter et al. NM 2011). This can be shown with a simple thought experiment: Assume all target assays in a spectral library will deliver true identifications because they all deliver high S/N. The scoring function (discriminant score) cannot completely separate the score distributions from true and false (decoy) signals. In this case the FDR estimate described above will produce a cutoff that is too conservative (depending on the level of overlap between the distributions). Unlike in spectrum-centric analysis false identifications do not uniformly distribute over target and decoy search space. This estimate is however an estimate that is proportional to the true FDR and hence, suitable for benchmarking. For this reason I strongly recommend to rename the "Effective FDR" to "Proxy FDR" or similar.

Response to Reviewers' Comments

Response to Reviewers

Reviewer: 1

Comments:

My issues with the original manuscript have been well addressed, and I suggest acceptance with an additional minor comment.

1. I don't think the heading "Identification of PTM peptides with DreamDIA" is fully appropriate since this section only reports the numbers of identified deamidated peptides. Other PTMs are not tested, and the most important task for PTM analysis – site localization – is not involved in this section. The reason that DreamDIA identified more deamidation peptides is not discussed, either. But this is not the subject of the manuscript, maybe the topic in future investigations, and should not prevent publication of this manuscript. I suggest changing the heading to "Identification of more deamidated peptides with DreamDIA" and move this section to the Supplementary Information.

Authors' response: We thank the reviewer for this suggestion. In the revised manuscript, we have modified the heading to "Identification of more deamidated peptides with DreamDIA" and have moved this section to Supplementary Note 1.

Reviewer: 2

Comments:

The authors have sufficiently addressed my previous comments and greatly improved their manuscript. There are a few grammar mistakes and typos that I am confident will be addressed in the final version.

1. Please define 'RT cycles'. (I think 'acquisition cycle' would be the more common term?)

Authors' response: We thank the reviewer for this advice. "Acquisition cycle" is indeed more precise here compared with "RT cycles", so we have changed to this term in the revised manuscript.

2. I was wondering whether an experimentally measured mass signal could be (accidentally) assigned to multiple precursors given the large number of transitions considered for each precursor. Could the authors please comment on this?

Authors' response: It is true that some signals might be assigned to multiple precursors due to the intrinsic peptide co-elution characteristic of DIA. However, it doesn't affect the efficacy of the proposed methods. Instead, although it is highly likely that some fragment ions are misaligned to multiple precursors, it is nearly impossible that all the fragment ion signals of a true precursor are from misalignment. Therefore, there will be distinguishable patterns between signals from true and

false precursors when a large number of transitions are considered, which could then be learnt by the DNN model of DreamDIA for better precursor identification.

Reviewer: 3

Comments:

The manuscript has improved significantly in clarity and depth of analysis. The authors have addressed all my comments. There is one final concern to be addressed:

1. page 6, "Effective FDR at precursor level was calculated as the number of Arabidopsis precursors identified divided by the number of all precursors identified." This is the FDR estimation procedure as described in literature to be suitable for spectrum-centric analysis (Elias and Gygi Nature Methods 2007). It is not an accurate estimate for FDR in peptide-centric analysis (Reiter et al. NM 2011). This can be shown with a simple thought experiment: Assume all target assays in a spectral library will deliver true identifications because they all deliver high S/N. The scoring function (discriminant score) cannot completely separate the score distributions from true and false (decoy) signals. In this case the FDR estimate described above will produce a cutoff that is too conservative (depending on the level of overlap between the distributions). Unlike in spectrum-centric analysis false identifications do not uniformly distribute over target and decoy search space. This estimate is however an estimate that is proportional to the true FDR and hence, suitable for benchmarking. For this reason I strongly recommend to rename the "Effective FDR" to "Proxy FDR" or similar.

Authors' response: We thank the reviewer for the detailed explanation of the difference of FDR estimation methods between peptide-centric analysis and spectrum-centric analysis. We totally agree with the reviewer and have used the term "Proxy FDR" in the revised manuscript instead of the "effective FDR".